# Phytochemicals and Inflammation: Is Bitter Better?

**DOI:** 10.3390/plants11212991

**Published:** 2022-11-06

**Authors:** Dorin Dragoș, Madalina Petran, Teodora-Cristiana Gradinaru, Marilena Gilca

**Affiliations:** 1Department of Medical Semiology, Faculty of Medicine, Carol Davila University of Medicine and Pharmacy, 020021 Bucharest, Romania; 21st Internal Medicine Clinic, University Emergency Hospital Bucharest, Carol Davila University of Medicine and Pharmacy, 050098 Bucharest, Romania; 3Department of Functional Sciences I/Biochemistry, Faculty of Medicine, Carol Davila University of Medicine and Pharmacy, 050474 Bucharest, Romania

**Keywords:** taste, bitter, sweet, sour, phytochemicals, anti-inflammatory

## Abstract

The taste of a herb influences its use in traditional medicine. A molecular basis for the taste-based patterns ruling the distribution of herbal (ethno) pharmacological activities may not be excluded. This study investigated the potential correlations between the anti-inflammatory activity (AIA) and the phytocompound taste and/or its chemical class. The study relies on information gathered by an extensive literature (articles, books, databases) search and made public as PlantMolecularTasteDB. Out of a total of 1527 phytotastants with reliably documented taste and structure available in PlantMolecularTasteDB, 592 (for each of which at least 40 hits were found on PubMed searches) were included in the statistical analysis. A list of 1836 putative molecular targets of these phytotastants was afterwards generated with SwissTargetPrediction tool. These targets were systematically evaluated for their potential role in inflammation using an international databases search. The correlations between phytochemical taste and AIA, between chemical class and AIA, and between the taste and the number of inflammation related targets were statistically analyzed. Phytochemical taste may be a better predictor of AIA than the chemical class. Bitter phytocompounds have a higher probability of exerting AIA when compared with otherwise phytotastants. Moreover, bitter phytotastants act upon more inflammation related targets than non-bitter tasting compounds.

## 1. Introduction

Herbs have been employed in traditional medicine to treat various ailments via traditional practices [1]. The taste of the herbs as an ethnopharmacological descriptor is one of the factors indicating/used to determine their therapeutic activities [2,3,4,5].

Anthropologists tried to explain this approach through the model of sensory ecology, which focuses on the flow of sensory information between organisms and how this is used for survival. According to this model, sensation mediates interactions among plants and humans [6]. The similarity in the sensorial effect/impact of two different herbs indicates similarity in their interactions with human biology/pathology—therefore plants with similar organoleptic properties are expected to have similar therapeutic activities. Does this model have a biological basis? Since the discovery of the widespread location of extraoral taste receptors and their functions [7,8,9], the scientists started to catch a glimpse on a potential answer to this question [10,11]. More than that, taste receptors and other chemosensors were proposed as broad ecological sensors important in inflammation and many other biological phenomena, being located in many “strategic” systems and organs linked to this pathological process [11]. Regarding the hypothesis of a taste-based determinism of the pharmacological activity, we focused our research on a specific biological activity of plants and phytochemicals: the anti-inflammatory activity. The story of aspirin, which was derived from willow bark salicylates, is a good example of the huge therapeutic potential of medicinal plants: aspirin discovery represents a milestone in the history of pharmaceutical industry, representing one of the prototypes of the synthetic non-steroidal anti-inflammatory drugs. Moreover, the aspirin story continues today: it is still currently the most widely used drug worldwide [12]. With the emerging new viral infections that mediate hyperinflammatory responses, analyzing natural resources, in order to identify new anti-inflammatory compounds that may limit the host tissue injury, may offer alternative therapeutic or preventive solutions [13,14].

In this paper the term phytotastant designates a plant-derived taste active compound. For simplicity, we included under this generic term, also the phytocompounds able to induce astringency or pungency.

In order to study the role of herbal taste in the distribution of ethnopharmacological activities, we introduced the concept of phytomolecular taste, or plant molecular taste, in an earlier paper, as derived from the inseparable connection (*samavaya*) postulated in Ayurveda between taste (*rasa*) and substance (*dravya*), and it is justified by the fact that the ethnopharmacological activities of medicinal plants seem to be better predicted by the taste of the major constituent phytocompounds than by their phytochemical class [15]. The phytomolecular taste, or the plant molecular taste, represents the virtual taste profile resulted from the contribution of all the major phytotastants found in a medicinal plant [15,16]. In a previous study we found that bitter phytomolecular taste of Indian herbals is statistically associated with ayurvedic anti-inflammatory activity (Sanskrit sothahara) [15]. The question that immediately arises is whether this association is still valid for evidence based anti-inflammatory activity. A preliminary statistical analysis performed on 466 taste active phytochemicals confirmed this hypothesis [17], and using an online tool (SwissTargetPrediction, available at http://www.swisstargetprediction.ch/ accessed on 25 September 2022) [18] for all the phytotastants available in BitterDB [19,20] and SuperSweetDB [21], we found that bitter phytotastants (by comparison with sweet ones) have a higher probability of targeting molecules involved in anti-inflammatory pathways [22]. These partial results encouraged us to develop a larger database of phytotastants (www.plantmoleculartastedb.org accessed on 21 September 2022) as a platform that would allow us to perform a stronger statistical analysis of this potential link.

The general aim of the study was to explore whether the anti-inflammatory activity (AIA) may be predicted by the taste of/orosensations generated by the phytocompounds or by their chemical class. The specific objectives of this research were to: (1) evaluate the potential concordance between the taste of/orosensations generated by phytochemicals and the AIA; (2) evaluate the potential concordance among the chemical classes of the phytocompounds and the AIA; (3) determine which attribute of the phytocompounds (either the chemical class or the taste) predicts better AIA; (4) explore whether the putative association between the chemical class of the phytocompounds and their AIA is mediated by the taste of the phytocompounds; (5) explore the putative association between the taste of/orosensations generated by the phytocompounds and their interaction with inflammation-related targets; and (6) propose explanations for observed associations (if they exist).

We have considered in our study seven tastes/orosensations (astringent, bitter, pungent, salty, sour, sweet, and umami), which were derived from integrating traditional medicine (Ayurveda and Traditional Chinese Medicine) with modern taste sciences. The taste of the medicinal plants was a fundamental descriptor and an important indicator of the ethnopharmacological activity in both Traditional Chinese [23] and Ayurvedic [24] medicine, which are, at a global level, the most ancient, yet living, herbal traditions [25]. More often than not taste by itself was considered to have intrinsic therapeutic potential [10,26]. There are six tastes/orosensations in Ayurvedic medicine: astringent, bitter, pungent, salty, sour, sweet [24,27] and five tastes/orosensations in Traditional Chinese Medicine: bitter, pungent, salty, sour, sweet [23]. As for umami, it is one of the five basic taste modalities recognized by the modern science (sweet, bitter, salty, sour, umami), with distinct taste receptors on the tongue [28], as well as in non-taste related organs [29,30,31]. Hence the seven tastes/orosensations we have included in our analysis.

## 2. Results

The total number of phytocompounds included in the statistical analysis was 592: 452 with proven AIA, 140 without AIA.

Among the tastes/orosensations included in our study (astringent, bitter, pungent, salty, sour, sweet, umami), only bitter taste had a statistically significant positive correlation with AIA, as reflected by the greater-than-one odds ratio (OR) with a 95% confidence interval not including one. By contrast, sour and sweet have a negative association with AIA, as reflected by the less-than-one OR with a 95% confidence interval not including one (Table 1). Surprisingly, pungent seemed to have no association with AIA. However, an analysis performed only on pungent-not-bitter substances revealed an inverse relationship with AIA (Table 1).

Among the chemical classes, only the flavonoids have a statistically significant positive correlation with AIA, as reflected by the greater-than-one OR with a 95% confidence interval not including one. By contrast, the saccharides and the short aliphatic acids have a negative association with AIA, as reflected by the less-than-one OR with a 95% confidence interval not including one (Table 2).

In order to determine whether the positive correlation between flavonoids and AIA was mediated by the bitter taste, the analysis was restricted to the flavonoids (see the last but one row in Table 1). A similar computation was done for flavonoid glycosides (see the last row in Table 1).

For short aliphatic acids, sour taste might be conceived as the mediator of the association with the absence of AIA—however, this cannot be proven statistically as all short aliphatic acids in our collection of phytocompounds are sour. Similarly, sweet taste cannot be statistically proven as the mediator of the association between saccharides and lack of AIA, as all but two saccharides (one with, the other without AIA) are sweet.

In total, 27 comparisons were performed. Therefore, according to Bonferroni correction, the corrected significance level should be 0.05 divided by 27 ≈ 0.002.

It should be noted that if Bonferroni correction is applied, the only statistically significant correlations are those between bitter taste and AIA (which is by far the statistically strongest correlation), between saccharides and lack of AIA, and between short aliphatic acids and lack of AIA. None of the correlations between chemical classes and AIA is statistically significant enough to withstand the rigors of Bonferroni correction. However, if we mollify our approach in order to get a more nuanced view, there are correlations between flavonoids and AIA and between flavonoid glycosides and AIA, and also between the sweet and sour tastes and the lack of AIA.

Swiss Target Prediction was successful for 548 (out of the total of 592) compounds, revealing a total of 1836 possible targets; out of each, 363 and 1245 did and did not have an inflammation related role, respectively (as revealed by the literature search). For the remaining 228 the literature search was inconclusive. Table 3, which reflects the correlation between the taste of phytocompounds and the number of AI effect linked targets, demonstrates highly statistically significant correlation for bitter taste (direct correlation) and for sour and sweet tastes (inverse correlations). More specifically, bitter tasting compounds act upon more inflammation-related targets than non-bitter tasting compounds (a median of 25 compared to a median of 20). Conversely, sour tasting compounds act upon less inflammation-related targets than non-sour tasting compounds (a median of 15 compared to a median of 24); the same holds for sweet tasting compounds (a median of 19 compared to a median of 25).

## 3. Discussion

The present statistical study included 592 phytotastants available in PlantMolecularTasteDB (http://plantmoleculartastedb.org), for each of which at least 40 hits were found on PubMed searches. The results obtained by means of Chi-squared test or Fisher’s exact test showed that the taste of a phytocompound may be a better predictor of its anti-inflammatory activity than its chemical class, and that bitter phytotastants have a higher probability of exerting anti-inflammatory activity when compared with otherwise tasting phytochemicals.

### 3.1. For a Given Phytochemical, Is the Taste Responsible for the Anti-Inflammatory Activity or the Lack Thereof?

As both bitter taste and flavonoid class are associated with AIA, the supposition arises that bitter taste might be instrumental in the flavonoid-AIA association—indeed, the high OR (4.27) suggests that bitter taste might be a predictor/indicator of the potential for AIA of a flavonoid, but the result does not reach the necessary level of statistical significance. However, this does not hold for flavonoid glycosides, as the corresponding OR is almost equal to unity. It is not our intention to deny that the chemical class is relevant for the pharmacological activity of a phytocompound (particularly for AIA). We only state, based on the evidence provided by our study, that taste is even more relevant—as bitter taste proved to be more closely associated with AIA than a given chemical class such as flavonoids. Therefore, one may speculate that the AIA of a given class is largely dependent on its ability to activate bitter taste receptors. Indeed, experts in taste sciences have already underlined that polyphenols, among which flavonoids represent the biggest class, bind to the oral and extraoral bitter taste receptors TAS2Rs, which modulate the signaling pathways involved in anti-inflammatory processes, with consequent health beneficial effects [32]. Our present results support their conclusions.

Interestingly, in a previous study on the chemical space of natural bitterants, scientists found that the chemical superclasses of compounds are not relevant for the promiscuity and selectivity for TAS2Rs [33]. Moreover, bitter compounds with a high degree of chemical structural similarity (e.g., caffeine and theobromine) may have different promiscuity profiles [33]. According to their results, and our results as well, the chemical structural characteristics seem not to be sufficient, at least in some cases, for the complete description of the pharmacological activity of natural compounds (e.g., interaction with various biological targets, which triggers a biological response).

The negative association of sweet taste with AIA found in our study is not unexpected, since previous studies have shown that high sugar intake (e.g., sucrose, fructose) is associated with chronic inflammation [34,35,36,37]. Despite this trend, certain sweet phytocompounds (e.g., glycyrrhizin, brazzein) may exert anti-inflammatory activity [38,39,40].

As the lack of AIA is correlated with saccharides and short aliphatic acids among the chemical classes and with sweet and sour among the tastes, it is tempting to suppose that sweet taste mediates the correlation for saccharides and sour taste for short aliphatic acids. Nonetheless, this assumption could not be statistically endorsed.

The surprising lack of association between pungent and AIA is most probably the consequence of the associated bitter taste in many of the pungent compounds (32 out of 81), the positive association of bitter taste with AIA effacing the negative association of pungent orosensation with AIA. If the pungent-only compounds are analyzed separately, the inverse relationship between pungent and AIA is revealed.

We underline here that ”pungency” category in PMTDB is defined in accordance with Ayurveda concept of pungency (katu), which has a more versatile significance than the English “spicy/hot” category [15]. Pungency (katu) refers to pungent, sharp, penetrating, hot, caustic, acrid¸ but also to strong-scented, fragrant, exhaling strong odour, and stimulating (about smell) [41]. Since aromatic plants, rich in essential oils (even those inducing a cooling sensation, such as mint), are characterized as pungent (katu) in Ayurveda [42], we have used the “pungent” (katu) descriptor for those volatile compounds which induce cooling (menthol) [43], tingling (sanshool) [44], or numbing (eugenol) [45] sensations.

Pungency and other related somatosensory sensations, like hotness, were reported to be directly related to inflammation through the contribution of some TRP channels [46,47] that have a heterogenous and complex modulatory activity on the inflammatory response. The activation of some TRPs promotes inflammation (e.g., TRPA1) [46,47,48], while the activation of others attenuates inflammation (e.g., TRPM8) [49]; there is still a third category of TRPs, which have dual effects, either anti-inflammatory or proinflammatory depending on the inflammatory context (e.g., TRPV1) [47,50,51]. Moreover, there is a certain ambiguity in some cases so that scientists even concluded, regarding the agonists of certain TRPs, that whether they “promote or inhibit inflammation remains unclear” [52]. Activation followed by desensitization is a putative mechanism explaining the beneficial effect in inflammatory conditions of herbs containing TRPA1 agonists—TRPA1 was the most frequently reported target of the voice protecting plants, most of which contain natural TRPA1 agonists [53]. Some plant active principles deemed as pungent in Ayurvedic medicine, such as menthol and eugenol, have a dual activity, stimulating both TRPA1 (proinflammatory effect) and TRPM8 (anti-inflammatory effect) [54,55], thereby adding to the complexity of the inflammation-related effects of pungent phytocompounds.

### 3.2. Are Certain Bitter Taste Receptors Involved in the Anti-Inflammatory Activity?

We had not enough data to perform the statistical analysis necessary for answering this question, but increasing experimental evidence shows that the AIA of bitter phytocompounds from various chemical classes is at least partially mediated by bitter taste receptors TAS2R [56,57,58]. Unfortunately, these are mainly indirect proofs of TAS2Rs’ involvement because the majority of the studies use either non-specific TAS2Rs agonists, or evaluate only their effects on the release of proinflammatory cytokines or on the inhibition of mast cell activation, not excluding other TAS2R-independent potential anti-inflammatory mechanisms activated by the respective phytotastants. One of the best ways to obtain direct evidence on TAS2Rs involvement relies on TAS2R knockdown by siRNA silencing. Tiroch et al. have recently used a CRISPR-Cas9-edited TAS2R50ko cell model of human gingival fibroblasts (HGF), which revealed the involvement of TAS2R50 in the resveratrol-induced reduction in the LPS-evoked IL-6 release [59]. In another study, a similar method was used to show that TAS2R16 activation by its specific agonist salicin mediated inflammation resolution in HGFs, by antagonizing NF-kB signaling via NF-κB p65 nuclear translocation and by suppressing LPS-induced expression of pro-inflammatory cytokines, including IL-6 and IL-8 [60]. Another type of direct evidence on the anti-inflammatory activity of TAS2Rs agonists is derived from studies on TAS2Rs with several haplotypes. For instance, TAS2R38 has two frequent haplotypes: TAS2R38-PAV (functional), and TAS2R38-AVI (non-functional). Allyl isothiocyanate, an agonist of TAS2R38 [61], displayed an anti-inflammatory activity in vitro, which was dependent on TAS2R38 receptor functionality [62].

Scientists also suggested a close structural similarity between the TAS2R14 binding site residues and the active sites of COX-2, since all studied NSAIDs have affinity for TAS2R14 and inhibit COX [63]. Therefore, it would be reasonable to assume that certain molecular fingerprints or structural scaffolds are shared by TAS2Rs and other molecular targets involved in anti-inflammatory pathways (e.g., enzymes, receptors, channels), and this may represent the biological basis for a taste-based approach to predicting the therapeutic potential of various (ethno)pharmacological agents.

The present study has several limitations: 1. the relatively low number of phytotastants included in the study, which is a consequence mainly of the relative paucity of reliable data regarding the taste of the various phytocompounds and of the limited number of phytocompounds for which reliable studies are available regarding their antiinflammatory activity; and 2. The information regarding the molecular targets generated with Swiss Target Prediction webtool did not allow us to differentiate the type of putative ligand-target interaction (activation or inhibition).

## 4. Materials and Methods

The study relies on the information gathered and made public in PMTDB (http://plantmoleculartastedb.org), which contains a total of 1527 phytocompounds from traditionally used herbs. According to our knowledge, PMTDB is the largest database dedicated to phytotastants and orosensation active phytochemicals [16]. The existence of well-documented, reliable information about the taste of the phytocompound was the criterion for including a phytocompound in this database.

For any given compound, AIA is one of the most, and implicitly one of the first, studied biological activities, only second to antimicrobial and anticancer activities, according to our search in PubMed performed on the 13 February 2022 (Table 4).

The criterion for including a phytochemical in the statistical analysis was that the PubMed search using the name of the phytotastant as keyword produced at least 40 articles (the inclusion criterion).

Anti-inflammatory activity was considered evidence-based if supported by at least one correctly conducted study, irrespective of being performed in vitro or on animal or human subjects. However, among the phytochemicals with positive evidence for AIA, only those that also fulfilled the inclusion criterion were included in the statistical analysis.

Taking into account that lack of AIA is difficult to identify by searching the literature, we considered as “negative evidence for AIA” the fulfilment of two criteria: 1. lack of any positive evidence at the search performed using the combination “[specific phytochemical name] AND anti-inflammatory OR anti-inflammatory OR inflammation”; and 2. the PubMed search using the name of the phytotastant as keyword produced at least 40 articles, meaning that the phytocompound has been the object of a sufficient number of studies for the AIA to be identified and that the lack of positive evidence is not caused by the lack of studies.

Only 592 phytotastants (either with negative or positive evidence of AIA) out of 1527 in PMTDB fulfilled the inclusion criterion and were therefore included in the statistical analysis (Figure 1).

Next, in order to validate the results of the first analysis, we performed a second analysis. SwissTargetPrediction (http://www.swisstargetprediction.ch/ accessed on 25 September 2022), was employed to determine the putative molecular targets of the compounds that were included in the previous statistical analysis. SwissTargetPrediction is a webtool developed by the Swiss Institute of Bioinformatics to perform ligand-based target prediction for any bioactive small molecule [18]; therefore, the phytochemicals with molecules that are too large, with more than 200 characters per SMILES (e.g., acemannan, inuline, tannic acid) or proteins (e.g., brazzein, thaumatin) could not be submitted to SwissTargetPrediction tool. Similarly, for those with less than 5 heavy atoms (e.g., acetic acid, formic acid), the list of targets could not be generated with SwissTargetPrediction. Therefore, we were left with 548 compounds (out of the original 592).

A literature search was subsequently performed to establish whether these molecular targets of the phytochemicals may play a role in the inflammatory response. Several databases (https://www.uniprot.org accessed on [64], https://www.proteinatlas.org accessed on 18 September 2022 [65], https://pubchem.ncbi.nlm.nih.gov accessed on 22 September 2022 [66], https://www.ncbi.nlm.nih.gov/protein accessed on 19 September 2022 [67]) were interrogated systematically on each entry corresponding to a specific molecular target from our list, using as keywords all the words that contain “inflammat” (inflammatory, anti-inflammatory/antiinflammatory, inflammation, pro-inflammatory/proinflammatory, etc.).

The correlations between a categorical parameter (such as taste) and a continuous parameter (such as the number of inflammation related targets) were analyzed by means of Mann–Whitney test.

The correlations between two categorical parameters (such as taste and AIA or chemical class and AIA) were analyzed by means of Chi-squared test or Fisher’s exact test (the latter when at least one expected value in the Chi-squared test was less than 5). The Chi-square test was performed on 2 × 2 tables built on the model in Table 5.

The results were considered statistically significant if the *p*-value was below the generally accepted threshold of 0.05. When multiple comparisons were performed, the significance level (commonly set at 0.05) was lowered according to Bonferroni correction: the corrected significance level was 0.05 divided by the number of comparisons [68]. All the statistical calculations were performed using the R language and environment for statistical computing and graphics, version 4.0.3 (copyrighted by The R Foundation for Statistical Computing).

In order to perform Chi-square test, the categories should be disjunct [69] (i.e., each compound should fall in one, and only one, chemical class), therefore we had to classify the phytocompounds in mutually exclusive chemical classes. About half of these classes (alkaloid glycosides, alkylamides, anthraquinone glycosides, anthraquinones, carotenoids, coumarin glycosides, diterpenoid glycosides, fatty acid esters, fatty acids, fatty alcohols, phenylpropanoid glycosides, polyketides, proteins, steroids, sulfur compound glycosides, tannin glycosides) were too sparsely populated (less than 6 phytocompounds each) to be included in the statistical analysis. Those included in the statistical analysis will be listed in the first column of Table 2.

## 5. Conclusions

The anti-inflammatory activity of plant-derived compounds is more strongly associated with their taste than with their chemical class. Bitter phytocompounds have a higher probability of playing an inflammation related role, exerting anti-inflammatory activity, by contrast to sour and sweet phytochemicals, which have a higher probability of being devoid thereof.

*Directions for future research*. If followed up by experimental assays, the present results are likely to furnish new insights to the connection between phytochemical tastes/orosensations and their role in the regulation of inflammation. The precise anti-inflammatory mechanisms of bitter phytochemicals should be investigated as soon as selective, potent TAS2R agonists and antagonists become available. Future studies based on TAS2Rs knockout experiments should also be dedicated to check the present findings. It would be interesting to further extend the analysis of the relationship between various categories of “pungency” or TRPs phyto-ligands and inflammation. Another future direction would be to develop a standard complete taste profile for each tastant, based on organoleptic evaluation and also on in vitro assays, which should be performed systematically on all the known human taste receptors and chemosensors involved in other gustative sensations. Organoleptic evaluation, used more in the past, has some limitations due to sensorial interindividual and intercultural variability, heterogeneity of protocols, high detection threshold of compounds, etc. [33,70,71]. Characterization of tastants based on in vitro assay with taste receptors is a better solution for detection of mild tastants, which are difficult to be identified using only sensory methods [33]. A standard complete taste profile of the phytochemicals will allow further refinements on the classification of tastants in relationship with their anti-inflammatory activity. It would be also interesting to develop on PMTDB online platform an interactive tool for prediction of the anti-inflammatory activity of (phyto)tastants, which may accelerate the discovery of new potent therapeutic agents for controlling inflammatory states.

## Figures and Tables

**Figure 1 plants-11-02991-f001:**
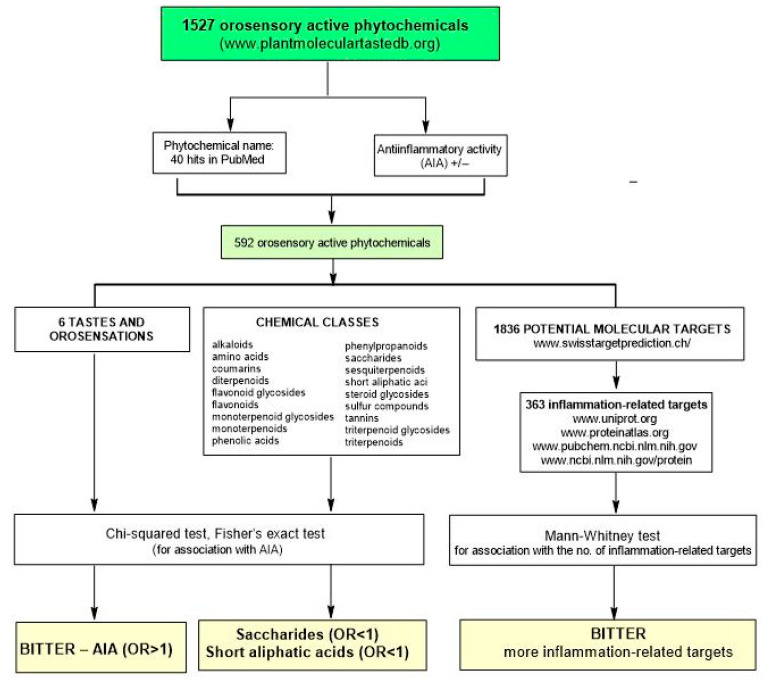
Flowchart outlining the steps of data preparation and statistical analysis.

**Table 1 plants-11-02991-t001:** The correlation between anti-inflammatory activity and taste. Legend: AIA—anti-inflammatory activity; N/A—not applicable; Note. The second column contains the number of phytocompound in each of the 4 cells of the 2 × 2 tables used for performing Chi-square test and/or Fisher’s exact test: a = taste + AIA+, b = taste + AIA−, c = taste − AIA+, d = taste − AIA−. Statistically significant results are bold typed.

Taste	Counts (a, b, c, d)	Chi-Square Statistics	*p*-Value by Chi-Square	*p*-Value by Fisher’s Exact Test	Odds Ratio (OR)	95% Confidence Interval for OR
astringent	(73, 14, 379, 126)	2.7535	0.097	0.077	1.73	0.93–3.44
bitter	(322, 68, 130, 72)	23.435	**1.3 × 10^−6^**	**1.4-06**	**2.62**	**1.74–3.94**
pungent	(58, 20, 394, 120)	0.090859	0.76	0.67	0.88	0.5–1.62
salty	(1, 2, 451, 138)	N/A	N/A	0.14	0.15	0.003–2.97
sour	(24, 18, 428, 122)	8.1283	**0.004**	**0.0042**	**0.38**	**0.19–0.77**
sweet	(62, 32, 390, 108)	6.019	**0.014**	**0.012**	**0.54**	**0.33–0.9**
umami	(4, 2, 448, 138)	N/A	N/A	0.63	0.62	0.09–6.89
pungent not-bitter	(13, 52, 38, 42)	11.8943	0.00056	0.0008	0.28	0.13–0.58
bitter (flavonoids only)	(53, 4, 6, 2)	N/A	N/A	0.16	4.27	0.32–38.32
bitter (flavonoid glycosides only)	(33, 4, 16, 2)	N/A	N/A	1	1.09	0.09–8.56

**Table 2 plants-11-02991-t002:** The correlation between anti-inflammatory activity and chemical class. Legend: AIA—anti-inflammatory activity, chemClass—chemical class; N/A—not applicable; Note. The second column contains the number of phytocompound in each of the 4 cells of the 2 × 2 tables used for performing Chi-square test and/or Fisher’s exact test: a = chemClass + AIA+, b = chemClass + AIA−, c = chemClass − AIA+, d = chemClass − AIA−. Statistically significant results are bold typed.

Chemical Class	Counts (a, b, c, d)	Chi-Square Statistics	*p*-Value by Chi-Square	*p*-Value by Fisher’s Exact Test	Odds Ratio (OR)	95% Confidence Interval for OR
alkaloids	(69, 31, 383, 109)	3.1281	0.077	0.07	0.63	0.39–1.06
amino acids	(22, 6, 430, 134)	0.003071	0.96	1	1.14	0.44–3.52
coumarins	(9, 2, 443, 138)	N/A	N/A	1	1.4	0.29–13.48
diterpenoids	(12, 1, 440, 139)	N/A	N/A	0.32	3.79	0.55–163.1
flavonoid glycosides	(51, 6, 401, 134)	5.2378	**0.022**	**0.013**	**2.84**	**1.18–8.27**
flavonoids	(59, 6, 393, 134)	7.5332	**0.0061**	**0.003**	**3.35**	**1.41–9.7**
monoterpenoid glycosides	(12, 1, 440, 139)	N/A	N/A	0.32	3.79	0.55–163.1
monoterpenoids	(32, 5, 420, 135)	1.6864	0.19	0.16	2.06	0.77–6.89
phenolic acids	(10, 3, 442, 137)	N/A	N/A	1	1.03	0.26–5.92
Phenylpropanoids *	(12, 2, 440, 138)	N/A	N/A	0.54	1.88	0.41–17.5
saccharides	(14, 16, 438, 124)	13.739	**0.00021**	**0.0003**	**0.25**	**0.11–0.56**
sesquiterpenoids	(19, 4, 433, 136)	0.22099	0.64	0.62	1.49	0.48–6.13
short aliphatic acids	(3, 7, 449, 133)	N/A	N/A	**0.0023**	**0.13**	**0.02–0.57**
steroid glycosides	(7, 5, 445, 135)	N/A	N/A	0.17	0.43	0.11–1.73
sulfur compounds	(9, 6, 443, 134)	N/A	N/A	0.14	0.45	0.14–1.58
tannins	(13, 1, 439, 139)	N/A	N/A	0.21	4.11	0.61–176.05
triterpenoid glycosides	(7, 2, 445, 138)	N/A	N/A	1	1.09	0.2–10.83
triterpenoids	(19, 4, 433, 136)	0.22099	0.64	0.62	1.49	0.48–6.13

* Others than the main classes of (iso)flavonoids, coumarins, stilbenes, lignans, and hydroxycinnamic acids.

**Table 3 plants-11-02991-t003:** Correlation between the count of inflammation-related targets and the taste as revealed by Mann–Whitney test. Q1, q3 = first and third quartiles. median [q1–q3] for yes/no = median [q1–q3] for the compounds with/without the corresponding taste.

Taste	Median [q1–q3] for Yes	Median [q1–q3] for No	W Statistics	*p*-Value
bitter	25 [21–28]	20 [14–26]	19917	3 × 10^−8^
astringent	23.5 [19–27]	24 [20–28]	20372.5	0.2
salty	16.5 [12.25–20.75]	24 [20–28]	2565	0.02
pungent	23.5 [20–28.75]	24 [19.5–28]	17727.5	0.6
sour	15 [6–21]	24 [20–28]	20315.5	3 × 10^−14^
Sweet	19 [14–24]	25 [21–29]	31957.5	4–12

**Table 4 plants-11-02991-t004:** Number of articles found in PubMed at the search on various biological activities of phytochemicals.

Biological Activity	A. Number of Articles Found in PubMed at the Search: Name of the Biological Activity and Phytochemical	Year of the First Published Paper for the Search A	B. Number of Articles Found in PubMed at the Search: Name of the Biological Activity	Year of the First Published Paper for the Search B
Antimicrobial OR anti-microbial OR antiinfectious OR anti-infectious OR antiinfective OR infection preventing OR antibacterial OR antiviral OR antiparasitic OR antifungal	8363	1946	2,506,227	1802
Anticancer OR antineoplastic OR antiproliferative OR cancer preventing	8032	1954	1,480,830	1934
**Anti-inflammatory OR anti-inflammatory OR antiphlogistic OR inflammation preventing**	**6449**	**1952**	**756,116**	**1826**
Antihypertensive OR hypotensive	640	1959	379,075	1913
Cardiotonic OR “heart tonic”	290	1954	215,217	1945
Diuretic	283	1956	105,834	1807
Immunostimulant OR immune enhancer	279	1972	429,982	1899
Choleretic	233	1972	30,051	1945
Cholagogue	230	1972	29,742	1852
Antacid	12	1969	27,690	1931

**Table 5 plants-11-02991-t005:** The 2 × 2 table used for performing Chi-square test (AIA = anti-inflammatory activity).

	AIA	No AIA
Taste/Chemical class	a	b
Not Taste/Chemical class	c	d

## Data Availability

The data used in this study are freely available in PMTDB (http://plantmoleculartastedb.org, accessed on 21 September 2022).

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
