# Peer review of "Phytochemicals and Inflammation: Is Bitter Better?"

_plants, 2022, doi:10.3390/plants11212991_

Round 1

Reviewer 1 Report

The paper is well written and sound. The statistical approach to the relationships between taste and pharmacological activity is appropriate and shows new and interesting observations which could hardly be obtained with different approaches. It could be somehow unexpected the result about pungent tastants; in fact, pungency and other somatosensory sensations like hotness are quite directly related to inflammation through the well known activation of TRP channels. Maybe the authors could add a few comment on this. 

A few recent paper should perhaps be cited in the discussion:

-on TRPA1pungent agonists and inflammation molecular targets:

Calcinoni O, et al. Herbs for Voice Database: Developing a Rational Approach to the Study of Herbal Remedies Used in Voice Care. J Voice. 2021 Sep;35(5):807.e33-807.e41. doi: 10.1016/j.jvoice.2019.12.027. Epub 2020 Jan 31. PMID: 32008898.

-on the relationship of bitterness and chemical space: 

Bayer S, et al. Chemoinformatics View on Bitter Taste Receptor Agonists in Food. J Agric Food Chem. 2021 Nov 24;69(46):13916-13924. doi: 10.1021/acs.jafc.1c05057. Epub 2021 Nov 11. PMID: 34762411; PMCID: PMC8630789.

Minor corrections:

-line 21: eliminate underline in "All these targets" in the abstract;

line 41: ...taste receptors and other chemosensors were proposed as  broad ecological sensors important in inflammation AND MANY OTHER BIOLOGICAL PHENOMENA...,

-LINE 46: bitter phytomolecular taste: this concept should be exlained since it is not self-evident for all readers

-line 72: among the taste orosensations...List here those chosen and explain the rational. Is the "pungent " descriptor (mediated by TRPA1) used in the same meaning of "hot" (mediated by TRPV1)? This point should be clearly  explained here and/or before table 1. 

Author Response

Reviewer 1

Comments and Suggestions for Authors

The paper is well written and sound. The statistical approach to the relationships between taste and pharmacological activity is appropriate and shows new and interesting observations which could hardly be obtained with different approaches.

> Thank for your kind words of appreciation

It could be somehow unexpected the result about pungent tastants; in fact, pungency and other somatosensory sensations like hotness are quite directly related to inflammation through the well known activation of TRP channels. Maybe the authors could add a few comment on this.

> Thank you for this eye-opening observation. Indeed, the lack of any association between pungent and anti-inflammatory activity (AIA) is rather surprising. The following additions to our article might shed a light on this issue:

“Surprisingly, pungent seemed to have no association with AIA. However, an analysis performed only on pungent not bitter substances revealed an inverse relationship with AIA (Table 1).” (in the Results section)

“The surprising lack of association between pungent and AIA is most probably the consequence of the associated bitter taste in many of the pungent compounds (32 out of 81), the positive association of bitter taste with AIA effacing the negative association of pungent orosensation with AIA. If the pungent only compounds are analyzed separately, the inverse relationship between pungent and AIA is revealed.” (in Discussions section 3.1)

A few recent paper should perhaps be cited in the discussion:

-on TRPA1 pungent agonists and inflammation molecular targets:

Calcinoni O, et al. Herbs for Voice Database: Developing a Rational Approach to the Study of Herbal Remedies Used in Voice Care. J Voice. 2021 Sep;35(5):807.e33-807.e41. doi: 10.1016/j.jvoice.2019.12.027. Epub 2020 Jan 31. PMID: 32008898.

> Done – the following text was added to the article (in Discussions section 3.1):

“Pungency and other related somatosensory sensations, like hotness were reported to be directly related to inflammation through the contribution of some TRP channels (Gouin et al., 2017; Nilius et al., 2007), which have a heterogenous and complex modulatory activity on the inflammatory response. The activation of some TRPs promotes inflammation (e.g. TRPA1) (Gouin et al., 2017; Kameda et al., 2019; Nilius et al., 2007), while the activation of others attenuates inflammation (e.g. TRPM8) (Ramachandran et al., 2013); there is still a third category of TRPs, which have dual effects, either anti-inflammatory or proinflammatory, depending on the inflammatory context (e.g. TRPV1) (Gouin et al., 2017; Silverman et al., 2020; Stampanoni Bassi et al., 2019). Moreover, there is a certain ambiguity in some cases, so that scientists even concluded, regarding the agonists of certain TRPs that whether they “promote or inhibit inflammation remains unclear” (Tsuji and Aono, 2012). Activation followed by desensitization is a putative mechanism explaining the beneficial effect in inflammatory conditions of herbs containing TRPA1 agonists – TRPA1 was the most frequently reported target of the voice protecting plants, most of which contain natural TRPA1 agonists (Calcinoni et al., 2021). Some plant active principles deemed as pungent in Ayurvedic medicine, such as menthol and eugenol, have a dual activity, stimulating both TRPA1 (proinflammatory effect), and TRPM8 (anti-inflammatory effect) (Bandell et al., 2004; Luo et al., 2019), thereby adding to the complexity of the inflammation-related effects of pungent phytocompounds.”

-on the relationship of bitterness and chemical space:

Bayer S, et al. Chemoinformatics View on Bitter Taste Receptor Agonists in Food. J Agric Food Chem. 2021 Nov 24;69(46):13916-13924. doi: 10.1021/acs.jafc.1c05057. Epub 2021 Nov 11. PMID: 34762411; PMCID: PMC8630789.

> Done – the following text was added to the article (Discussions section 3.1):

“Interestingly, in a previous study on the chemical space of natural bitterants, scientists found that the chemical superclasses of compounds are not relevant for the promiscuity and selectivity for TAS2Rs (Bayer et al., 2021). Also bitter compounds with a high degree of chemical structural similarity (e.g. caffeine and theobromine) may have different promiscuity profiles (Bayer et al., 2021). According to their results, and our results as well, the chemical structural characteristics seem not to be sufficient, at least in some cases, for the complete description of the pharmacological activity of natural compounds (e.g. interaction with various biological targets, which triggers a biological response). “

Minor corrections:

-line 21: eliminate underline in "All these targets" in the abstract;

> Done. Thank you for this observation!

line 41: ...taste receptors and other chemosensors were proposed as  broad ecological sensors important in inflammation AND MANY OTHER BIOLOGICAL PHENOMENA...,

> Done. Thank you for suggesting this addition!

-LINE 46: bitter phytomolecular taste: this concept should be explained since it is not self-evident for all readers

> Thank you for pointing out this shortcoming! The following text was added to the article (in Introduction):

“In order to study the role of herbal taste in the distribution of ethnopharmacological activities, we introduced the concept of phytomolecular taste or plant molecular taste in an earlier paper, as derived from the inseparable connection (samavaya) postulated in Ayurveda between taste (rasa) and substance (dravya) and it is justified by the fact that the ethnopharmacological activities of medicinal plants seems to be better predicted by the taste of the major constituent phytocompounds than by their phytochemical class (Dragos and Gilca, 2018). The phytomolecular taste or the plant molecular taste represents the virtual taste profile resulted from the contribution of all the major phytotastants found in a medicinal plant (Dragos and Gilca, 2018; Gradinaru et al., 2022).”

-line 72: among the taste orosensations...List here those chosen

> Thank you for this suggestion! The following text was added at the beginning of the Results section:

“included in our study (astringent, bitter, pungent, salty, sour, sweet, umami)”

and explain the rational.

> Thank you for this suggestion! The following text was added at the end of the Introduction section:

“We have considered in our study seven tastes/ orosensations (astringent, bitter, pungent, salty, sour, sweet, and umami), which were derived from integrating traditional medicine (Ayurveda and Traditional Chinese Medicine) with modern taste sciences. The taste of the medicinal plants was a fundamental descriptor and an important indicator of the ethnopharmacological activity in both Traditional Chinese (Bensky et al., 2004) and Ayurvedic (Sharma and Dash, 2006) medicine, which are, at a global level, the most ancient, yet living herbal traditions (Patwardhan et al., 2005). More often than not taste by itself was considered to have intrinsic therapeutic potential (Gilca and Dragos, 2017; Joshi et al., 2007). There a six tastes/orosensations in Ayurvedic medicine: astringent, bitter, pungent, salty, sour, sweet (Murthy, 1994; Sharma and Dash, 2006) and five tastes/orosensations in Traditional Chinese Medicine: bitter, pungent, salty, sour, sweet (Bensky et al., 2004). As for umami, it is one of the five basic taste modalities recognized by the modern science (sweet, bitter, salty, sour, umami), with distinct taste receptors on the tongue (Kurihara, 2015), as well as in non-taste related organs (Crowe et al., 2020; Liu et al., 2018; Shi et al., 2017). Hence the seven tastes/ orosensations we have included in our analysis.”

Is the "pungent " descriptor (mediated by TRPA1) used in the same meaning of "hot" (mediated by TRPV1)? This point should be clearly  explained here and/or before table 1.

> Thank you for this suggestion! The following text was added to the article (in Discussions section 3.1):

“We underline here that ”pungency” category in PMTDB is defined in accordance with Ayurveda concept of pungency (katu), which has a more versatile significance than the English “spicy/hot” category (Dragos and Gilca, 2018). Pungency (katu) refers to pungent, sharp, penetrating, hot, caustic, acrid¸ but also to strong-scented, fragrant, exhaling strong odour, and stimulating (about smell) (Monier-Williams, 2002). Since aromatic plants, rich in essential oils (even those inducing a cooling sensation, such as mint) are characterized as pungent (katu) in Ayurveda (Pandey, 2005), we have used the “pungent” (katu) descriptor for those volatile compounds, which induce cooling (menthol) (Peier et al., 2002), tingling (sanshool) (Kuroki et al., 2016), or numbing (eugenol) (Klein et al., 2013) sensations. “

Reviewer 2 Report

The question raised by DragoÅŸ et al. is of a broader interest to scientists and maybe even practitioners who deal with botanicals. I find the study well-done study; it is based on the database created by the authors earlier. Scientific approach and methodology as well as the limitations of the study are clearly communicated. The conclusions may appear a bit premature, however, the indication of the role of the TAS2R involvement deserves a closer look. The study may help to open new views on traditional knowledge as expressed by the use of certain bitter-tasting botanicals. 

To perfectionate the manuscript, I would recommend carrying out minor changes that I will indicate in the following.

1. Keep the abstract free of urls.

2. Define the term "phytotastant", as you mention it earlier in your database-related publication (Front Pharmacol. 2021; 12: 751712.). Since the term is new and there are - as far as I could find - no cross-references, it will be necessary to define it thoroughly.

3. Tables 1&2: It is not clear why certain entries are filled in bold font. Please explain their special relevance or leave the bold to the very few entries that you discuss in the text.

4. Discussion: There are sub-chapters 4.1. and 4.2. Shouldn't they be 3.1. and 3.2., respectively? Why at all assigning numbers here?

5. Line  189, a comma after "Cox-2" would help to structure the sentence.

6. p. 7: I'm not sure about the proper use of the term "orosensation", "orosensorially" since I believe it is more used for a false sensation of taste. Is this intended? Please correct otherwise by using a different term. 

Reviewer 3 Report

This article studied Phytochemicals and its anti inflammatory role detected on the basis of taste. In addition, the study examined the potential concordance among the chemical classes of the phytocompounds and the anti inflammatory activity. Before recommending this article for publication, there are some shortcomings for that should be resolve.

Abstract

Line 21 do not underline “All these targets”

The authors should provide how the data was collected. What was the criteria.

Introduction

Line 32 not only herbal taste but mostly herbal plants are considered as traditional medicines so use common sentence here “herbs have been considered in traditional medicine to treat various ailments via traditional practices”by citing related article https://doi.org/10.3390/molecules27196728

In the subsequent sentence the authors can write that “taste of the herb is one of the factor for indication or determination of the therapeutic activities”.

 Line 36-38 should be revise its not clear.

Overall introduction is well written but information about phytochemicals is limited.

The authors should add introduction of phytochemicals, its role in pharmaceutical industry, and more specifically anti inflammatory potential.

In discussion check the section numbers. These might be 3.1 and 3.2

Conclusion is well justified. The authors should discuss some points for the future studies molecular level studies are required to know about the involved mechanism and improvement of phytochemicals detection. 
